# Dual-Energy-CT for Osteitis and Fat Lesions in Axial Spondyloarthritis: How Feasible Is Low-Dose Scanning?

**DOI:** 10.3390/diagnostics13040776

**Published:** 2023-02-18

**Authors:** Dominik Deppe, Katharina Ziegeler, Kay Geert A. Hermann, Fabian Proft, Denis Poddubnyy, Felix Radny, Marcus R. Makowski, Maximilian Muhle, Torsten Diekhoff

**Affiliations:** 1Department of Radiology, Charité-Universitätsmedizin Berlin, Campus Mitte, Humboldt-Universität zu Berlin, Freie Universität Berlin, Charitéplatz 1, 10117 Berlin, Germany; 2Department of Gastroenterology, Infectiology and Rheumatology, Charité-Universitätsmedizin Berlin, Humboldt-Universität zu Berlin, Freie Universität Berlin, Charitéplatz 1, 10117 Berlin, Germany; 3Department of Radiology, University Hospital Rechts der Isar, Technical University Munich, Ismaninger Strasse 22, 81675 Munich, Germany

**Keywords:** dual-energy CT, axSpA, osteitis

## Abstract

Background: To assess the ability of low-dose dual-energy computed tomography (ld-DECT) virtual non-calcium (VNCa) images for detecting bone marrow pathologies of the sacroiliac joints (SIJs) in patients with axial spondyloarthritis (axSpA). *Material and Methods*: Sixty-eight patients with suspected or proven axSpA underwent ld-DECT and MRI of the SIJ. VNCa images were reconstructed from DECT data and scored for the presence of osteitis and fatty bone marrow deposition by two readers with different experience (beginner and expert). Diagnostic accuracy and correlation (Kohen’s k) with magnetic resonance imaging (MRI) as the reference standard were calculated for the overall and for each reader separately. Furthermore, quantitative analysis was performed using region-of-interest (ROI) analysis. *Results*: Twenty-eight patients were classified as positive for osteitis, 31 for fatty bone marrow deposition. DECT’s sensitivity (SE) and specificity (SP) were 73.3% and 44.4% for osteitis and 75% and 67.3% for fatty bone lesions, respectively. The expert reader achieved higher diagnostic accuracy for both osteitis (SE 93.33%; SP: 51.85%) and fatty bone marrow deposition (SE: 65%; SP: 77.55%) than the beginner (SE: 26.67%; SP: 70.37% for osteitis; SE: 60%; SP: 44.9% for fatty bone marrow deposition). Overall correlation with MRI was moderate (r = 0.25, *p* = 0.04) for osteitis and fatty bone marrow deposition (r = 0.25, *p* = 0.04). Fatty bone marrow attenuation in VNCa images (mean: −129.58 HU; ±103.61 HU) differed from normal bone marrow (mean: 118.84 HU, ±99.91 HU; *p* < 0.01) and from osteitis (mean: 172 HU, ±81.02 HU; *p* < 0.01) while osteitis did not differ significantly from normal bone marrow (*p* = 0.27). *Conclusion*: In our study, low-dose DECT failed to detect osteitis or fatty lesions in patients with suspected axSpA. Thus, we conclude that higher radiation might be needed for DECT−based bone marrow analysis.

## 1. Introduction

Bone marrow pathologies such as osteitis (bone marrow oedema) and fatty bone marrow lesions are important imaging findings of the sacroiliac joints (SIJs) in patients with axial spondyloarthritis (axSpA) [1,2]. Therefore, the Assessment of the SpondyloArthritis international Society (ASAS) emphasizes the detection of osteitis as a significant criterion for diagnosis of axSpA in its magnetic resonance imaging (MRI) classification system [3]. The diagnosis is further supported by structural changes such as erosions and fatty lesions, which have recently gained more attention for differentiating axSpA from other diseases [4,5]. 

While MRI is well suited to depict bone marrow pathologies, computed tomography (CT) is considered the gold standard for lesions of the calcified bone, i.e., erosion [6,7]. While conventional CT does not depict bone marrow changes, the recently introduced technique of dual-energy CT (DECT) virtual non-calcium (VNCa) reconstruction allows detection and assessment of bone marrow lesions [8]. Therefore, spectral CT imaging techniques have been increasingly used for musculoskeletal diagnosis [9]. DECT improves tissue characterization by acquiring CT data of the same volume at two different maximum tube voltage levels, thus characterized by two separate energy spectra. The spectral information is subsequently used in three-material decomposition for image postprocessing, elements such as calcium can be identified and subtracted from the image. The VNCa technique removes the calcified bone matrix of the spongious bone, thus providing CT images displaying only the bone marrow [10,11]. The ability of VNCa reconstructions to identify bone marrow pathologies was shown for acute spinal fractures [12,13,14], neoplasia [15], and osteitis in rheumatic diseases [16].

Therefore, DECT might combine the advantages of direct bone depiction for detection of structural lesions and identification of active inflammation. Furthermore, CT offers some additional advantages over MRI, for example, lower costs and shorter scanning times which is relevant for patients who cannot undergo MRI−scans due to back pain. Additionally, DECT might be an alternative imaging technique for patients with contraindications for MRI, e.g., pacemakers. However, DECT will use ionizing radiation for imaging.

Three studies investigating the detection of sacroiliitis reported sensitivities of 65.4–93% and specificities of 90–94.2% for DECT in patients with axSpA [17,18,19]. All three studies included quantitative measurements and proposed specific cut-off values for osteitis. However, the studies used rather high estimated radiation doses of 2 to 10 mSv, which is more than usually applied in the young age group of patients with axSpA. Furthermore, detection of fatty bone marrow lesions using spectral CT techniques has not yet been reported. While fat lesions are supposed to develop later in the course of the disease, they are specific imaging findings in a younger population and support the diagnosis of axSpA [20].

Therefore, the aim of our study was to assess the diagnostic accuracy of low-dose DECT virtual non-calcium (VNCa) images for detecting bone marrow pathologies of the SIJs, i.e., osteitis and fatty bone marrow deposition in patients with axSpA. We wanted to transfer earlier scientific findings to a clinical setting by investigating patients with various conditions.

## 2. Methods

### 2.1. Patients

Seventy-five patients with known or suspected axSpA from our hospital’s rheumatology department were considered for inclusion between February 2018 and November 2019. Included were patients with low back pain and clinically suspected axSpA as well as patients with known axSpA referred for a follow-up examination. Exclusion criteria were contraindications to MRI and/or CT, such as non-MR-compatible devices, pregnancy, and claustrophobia.

### 2.2. Imaging 

Patients underwent DECT and MRI on the same day. MRI was performed at 1.5 Tesla, and the protocol included a T1-weighted sequence (echo time (TE): 13 m; repetition time (TR): 441 ms; slice thickness: 4 mm; interslice gap: 0.4 mm; field of view: 300 mm; matrix: 512 × 526; acquisition time: 5:04 min) and short-tau inversion recovery (STIR) sequence (TE: 50 ms; TR: 3780 ms; inversion time (TI): 145 ms; slice thickness: 4 mm; interslice gap: 0.4 mm; field of view: 300 mm; matrix: 284 × 257; acquisition time: 7:24). Both pulse sequences were acquired in oblique coronal orientation.

DECT was performed as using a low-dose protocol for DECT on a 320-row single-source scanner (Canon Aquilion One, VisionToshiba Medical Systems, Otawara, Japan; installed in 2013). CT scans were acquired at 135 kV and 80 kV. 

Depending on clinical appearance, CT scans included the whole spine or the sacroiliac joints. The estimated effective dose was calculated using the dose-length product (DLP) and age- and sex-specific conversion factors for the pelvic scan field as recommended by the International Commission on Radiological Protection (IRCP) [21].

### 2.3. Image Postprocessing

From DECT raw data, 0.5 mm primary reconstructions were generated, which were further postprocessed using a proprietary software on the CT console (DE Image View, Version 6.0) to reconstruct 3 mm 120 kV-equivalent blended images and virtual non-calcium (VNCa) images. Both types of images were secondarily reconstructed in 3.0 mm oblique coronal planes to match the MRI datasets. 

### 2.4. Reader Training

Before the study, the two readers separately scored a set of 10 independent test cases that were not included in the final study population. Discrepancies were discussed by the readers in preparation of the study reading sessions. 

### 2.5. Image Reading

MRI and DECT images were separately anonymized and scored using Horos (The Horos Project, Version 3.3.6, Pureview, MD, USA) by two readers: reader 1 (TD), a musculoskeletal radiologist with 13 years of experience and reader 2 (DD), a research student with 3 years of experience in musculoskeletal imaging. The readers were blinded to identifying information and the images and results of the other imaging modality. 

For DECT, the readers had access to 120 kV-equivalent and VNCa images and were allowed to freely adjust the window levelling and to invert the images. For MRI, T1-weighted and STIR images were scored. Only images of the sacroiliac joints were shown, while spinal images were not included in the image stack. The two readers used an established 24-region scoring model of the SIJs, which divides each joint into 4 anterior, 4 middle, and 4 posterior quadrants [6]. Bone marrow oedema and fatty bone marrow deposition were scored as followed: 0: absent; 1: <33% of the quadrant, 33–66% of the quadrant; ≥66% of the quadrant. On VNCa images, osteitis was defined as a hyperdense subchondral lesion of the bone marrow adjacent to the cartilaginous joint surface. In contrast, fat lesions were defined as hypodense compared with normal bone marrow attenuation. Sclerosis was assessed on a three-point scale: 0: no sclerosis; 1: possible/little sclerosis; 2: marked sclerosis. Disagreements between the two readers were solved by two different experts: an expert with 7 years of experience in musculoskeletal imaging solved disagreements in the interpretation of DECT images and another expert with 18 years of experience in musculoskeletal imaging disagreements in MRI interpretation.

### 2.6. Quantitative Analysis

Based on the scoring results, Reader 2 performed a region-of-interest (ROI) measurement in each patient. ROIs were selected in regions with unequivocal osteitis, fat lesions, sclerosis, and normal bone marrow signal (measured in the center of the S2 vertebra) in T1-weighted images and STIR images as well as in 120 kv-eqivalent CT and in VNCa images. One ROI was placed for each lesion type and joint.

### 2.7. Data Analysis

Findings were analysed on a per-patient and per-point level. A patient/ joint was classified as positive for osteitis or fat lesions if the two readers agreed on its presence by assigning a score of two or higher. Sensitivity (SE), specificity (SP), and likelihood ratio were calculated for the two readers taken together and separately for each reader per joint and per patient. Correlation of DECT with MRI was calculated using Pearson’s r, while interrater reliability was assessed by calculating Kohen’s k. MRI served as the reference standard.

In a subpopulation analysis, joints with unequivocal sclerosis based on scoring were excluded from analysis, and sensitivity and specificity were calculated for the remaining cases on a per-joint and per-patient level for the two readers together and separately for each reader. 

An analysis separated by gender was performed.

Mean and standard deviation for quantitative measures were calculated and compared using Tukey’s multiple-comparison test for each lesion in each modality.

### 2.8. Ethics Approval and Informed Consent

The study was approved by the local ethics committee (EA1/086/16), and all patients gave written informed consent. Approval by the Federal Office for Radiation Protection was waived.

## 3. Results

### 3.1. Patients

Sixty-eight patients were included in the analysis. There were 40 patients with inflammatory diseases; among them, 35 patients were diagnosed with axSpA. A total of 12 patients were under medication with disease modifying anti-rheumatic drugs (DMARDS), 33 patients with non-steroidal anti-inflammatory drugs (NSAIDs) and two patients with corticosteroids. A total of 37 patients were positive for inflammatory back pain. A total of 20 patients reported regular physical activity. Exact data on the symptom duration was not available. Further details of patient inclusion and characteristics of the study population are presented in the flow chart in Figure 1.

### 3.2. Imaging 

The mean DLP for all scans, including 41 total spine scans, was 118.03 mGy*cm with a mean estimated effective dose of 1.51 mSv for pelvic scans. The mean CTDI was 27.54 mGy. 

### 3.3. Image Reading and Data Analysis

#### 3.3.1. Osteitis

Twenty-eight patients (30 joints) were scored as positive in MRI and 42 (56 joints) in DECT. Correlation of DECT with MRI was moderate on a per-patient level (r = 0.25, *p* = 0.04). Data on overall diagnostic accuracy and diagnostic accuracy for each reader are compiled in Table 1. Interrater reliability was poor for DECT (k = 0.19) and moderate for MRI (k = 0.56).

#### 3.3.2. Fatty Bone Marrow Lesions

Fatty bone marrow lesions were found in 31 patients in MRI (40 joints) and 33 patients (42 joints) in DECT. Correlation with MRI was moderate on both patient and joint levels (r = 0.25, *p* = 0.04; r = 0.37, *p* < 0.01). For diagnostic accuracy see Table 1. Interrater reliability was fair for DECT (k = 0.32) and moderate for MRI (k = 0.41). 

Examples of CT, T1-weightes images, STIR and DECT can be found in Figure 2.

Patient A. The 31-year-old male patient with axial spondyloarhtritis (axSpA) and severe osteitis of the sacral and iliac bones (white arrows). Bone marrow inflammation is indicated by high signal intensity in STIR (black arrow) and low T1 signal. In VNCa, osteitis is indicated by hyperdensity of the left sacroiliac joint (black arrow); however, the appearance does not allow estimating the extent of osteitis and is hard to distinguish from sclerosis (white arrow). Fatty bone marrow deposition can be seen with high signal intensity in T1 (black arrowhead) and can be detected in VNCa-Images on the right os sacrum.

Patient B. Fatty bone marrow deposition in a 46-year-old patient with axSpA. Fatty bone marrow deposition shows high T1 signal (white and black arrowhead). Standard CT does not visualize bone marrow changes, while they are apparent in reconstructions from dual-energy CT (DECT) by low HU (white/black arrowhead). Still, definitive distinction between normal and fatty bone marrow is not possible.

Patient C. Sacroiliac joint of a 35-year-old female with osteitis condensans and severe sclerosis of the anterior joint space (white arrow) surrounded by osteitis on the left side (white arrowhead). CT shows severe sclerosis of the sacral and iliac bones on the left (white arrow). The VNCa image does not allow reliable distinction between sclerosis (white arrow) and osteitis (white arrowhead) in this case.

### 3.4. Subpopulation Analysis

Subpopulation analysis was performed after exclusion of 71 joints and 26 patients with severe sclerosis. 

On the joint level, sensitivity decreased to 11.11% while specificity for osteitis increased to 90%. On the patient level, no osteitis was detected (with 75% specificity).

There were similar results if sclerosis were excluded for fat bone marrow deposition on joint level (SE: 50%; SP: 85.9%), while diagnostic accuracy increased on patients’ level (SE: 100 %; SP: 80.6 %). Further details are shown in Appendix A.

Furthermore, we performed a gender-specific analysis including 28 women and 40 men. Differences in the diagnostic performance between gender are shown in Appendix A.

### 3.5. Quantitative Analysis

ROI measurement was performed in 33 lesions with osteitis, 50 with fat metaplasia, and 35 with sclerosis as well as normal bone marrow in 43 patients. Results are shown in Figure 3.

Attenuation of fat metaplasia differed significantly from that of normal bone marrow (mean difference: −233.1 HU [−158.7–−307.5 HU; *p* < 0.01). For osteitis, the mean difference from normal bone marrow was 55.67 HU ([24.16–135.5 HU]; *p* = 0.27). Fat metaplasia and osteitis differed significantly with a mean difference of −288.8 HU ([−372.7–−204.9 HU]; *p* < 0.01). Osteitis and sclerosis showed a mean difference of 436.7 HU ([348–525.5 HU]; *p* < 0.01).

Fat metaplasia (mean difference: 124.5 HU [50.08–198.9 HU]; *p* < 0.01) and osteitis (mean difference: −91.86 HU [−171.7–−12.06]; *p* = 0.02) differed less markedly from normal bone marrow.

## 4. Discussion

In this study, we analysed ld-DECT for detecting bone marrow pathologies in a mixed study population of 68 patients in a clinical setting. While our results suggest that fair distinction of bone marrow lesions is possible when quantitative measurement is performed, analysis shows only limited diagnostic value for subjective scoring of reconstructed VNCa images by two readers. Osteitis and fatty lesions were detected with a moderate sensitivity (70.3–75%); however, specificity was limited for osteitis (44.4%) and for fatty lesions (63.3%). The more experienced reader in our study achieved higher diagnostic accuracy. Excluding sclerotic lesions did not improve detection of osteitis in our analysis. Quantitative measurement allows the differentiation of the two lesions from each other as well as differentiation of fat metaplasia from normal bone marrow. However, our results did not show confident distinction between osteitis and normal bone marrow.

Conversely, previous publications report high sensitivity and specificity for detecting osteitis on VNCa images [17,18,19]. However, these studies have some limitations regarding transfer of results to clinical practise. Wu et al. reported 87–93% sensitivity and 91–94% specificity for detection of osteitis of the SIJs but excluded osteitis in cases with obviously sclerotic articular surfaces, which is a common finding of the SIJs [17]. In addition, only lesions of the iliac bone were included. Other studies reported 81.3% sensitivity and 91.7% specificity and 90% sensitivity and 92.8% specificity including patients with subchondral sclerosis [18,19]. Despite training, the two readers in our study achieved lower performance in ld-DECT compared to high-dose DECT, which we attribute to the use of a different image acquisition protocol and postprocessing software. 

Detection of fatty bone marrow lesions with DECT has not been described in the literature so far. Fatty bone marrow lesions are typical imaging findings in patients with axSpA that are characterized by high signal intensity in T1-weighted MR images. They are assumed to represent replacement after inflammation and are associated with the occurrence of spinal syndesmophytes and fat metaplasia inside an erosion cavity of SIJs, which are findings that can corroborate the diagnosis [22,23]. However, we showed only moderate diagnostic accuracy for both the experienced and the less experienced reader. 

Published cut-off values for quantitative identification of osteitis range between −1.6 HU and −44 HU, with higher HU values in sacral bone and increasing attenuation in severe inflammation [18,19,24]. Our measurements were outside the reported range, which is attributable to several factors: (1) the low-dose protocol increases image noise, which is not well accounted for in our postprocessing method; (2) we applied a broader definition of lesions and measured multiple bones; (3) we also included patients with a different diagnosis than axSpA; and (4) we used different hardware and software in our analysis. 

Taken together, our results suggest that the CT technique investigated seems unsuitable for clinical use for detecting bone marrow pathologies. This may have three reasons. Firstly, we reduced the radiation exposure of DECT to low-dose levels. This may negatively impact DECT especially as it may render the energy spectra separation in the postprocessing more difficult, degrading proper bone marrow analysis. On the other hand, dose reduction is important in these relatively young patients with chronic diseases and the need for serial follow-up examinations. Previous studies used protocols with a rather high radiation dose, while we intended to reduce radiation exposure to a level acceptable in a routine clinical setting. The low-dose CT protocol used in our study reduced exposure to a mean estimated effective dose of 1.51 mSv, which corresponds to the dose of a conventional pelvic X-ray examination. Secondly, we investigated a mixed study population with almost half of the patients having noninflammatory conditions and only 28 patients with osteitis which lowers pretest probability. Thirdly, we included patients with severe sclerosis in our analysis, which can mimic osteitis in VNCa images, thus possibly reducing sensitivity and specificity alike [25]. To account for this fact in our analysis, we additionally performed a subpopulation analysis from which all patients with severe sclerosis were excluded. Still, diagnostic accuracy was not improved and did not reach the level reported by earlier investigators. Detection of fatty bone marrow lesions, however, benefited from exclusion of sclerosis, which resulted in perfect sensitivity (100%) and high specificity (82.6%) as sclerosis has high HU values while fatty bone marrow lesions have low HU values. However, greater experience can improve diagnostic performance in identifying both osteitis and fatty bone marrow lesions. Our analysis shows a higher diagnostic accuracy for the experienced reader on both the joint and patient levels.

Our study has some limitations. Two different readers resolved discrepancies between reader 1 and 2 for MRI and DECT. We investigated a mixed study population with various pathologies and only half of the patients had osteitis and fatty bone marrow lesions. Quantitative analysis was performed by a single reader. Due to ethical considerations, we were not able to compare Ld-DECT and DECT with higher exposure in this study. Therefore, several other influences and such as software versions and parameters can bias the comparison to the diagnostic accuracy published in other studies.

Still, MRI remains the gold standard for detection of inflammatory bone marrow lesions and DECT performed as low-dose CT could not achieve diagnostic accuracy of MRI. However, previous studies showed a good diagnostic performance for DECT using higher radiation and offer a possible alternative for patients with contraindications for MRI.

Our aim was to transfer earlier scientific findings to a clinical setting by investigating patients with various conditions, as would be encountered in clinical practice, by reducing radiation exposure and by analysing the performance of readers with different levels of experience. Exclusion of patients with sclerosis and quantitative measurements did not significantly improve diagnostic performance. In conclusion, ld-DECT of the SIJs in patients with axSpA did not allow sufficient detection of osteitis and bone marrow fat lesions in our study. The technique deserves further scrutiny to assess its possible clinical application. 

## Figures and Tables

**Figure 1 diagnostics-13-00776-f001:**
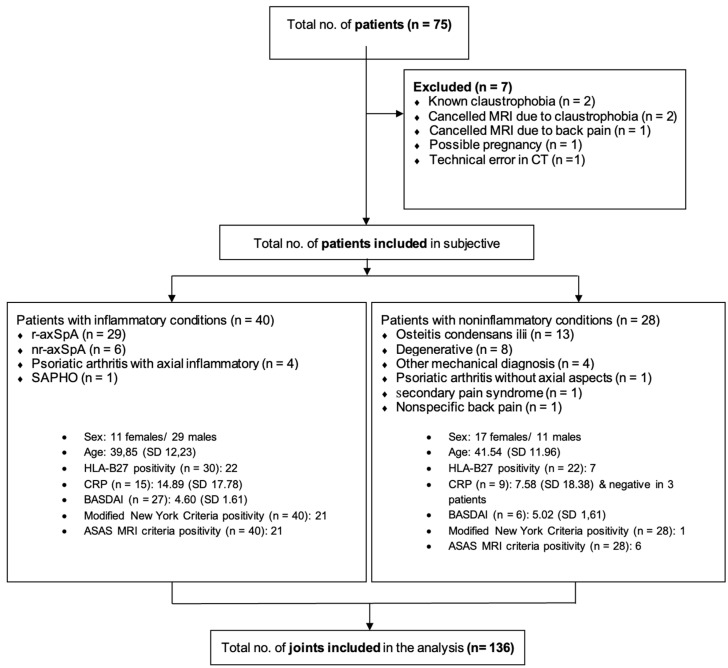
Flow chart of patient exclusion and characteristics of the final study population. A mixed study population of patients with inflammatory and noninflammatory diseases such as mechanical and degenerative diseases was investigated. A total of 35 patients were diagnosed with axial spondyloarthritis (axSpA). Final diagnoses were made by the treating rheumalogist. Females accounted for 41.2 % of the study population. Mean age was 40.54 years, and 27 patients were classified as positive according to the Assessment of SpondyloArthritis international Society (ASAS) definition of a positive MRI. r-axSpA, radiographic axial spondyloarthritis; nr-axSpA, nonradiographic axSpA; CRP, c-reactive protein; BASDAI, Bath Ankylosis Spondylitis Disease Activity Index. SAPHO, synovitis, acne, pustolisis, hyperostosis, and osteitis syndrome.

**Figure 2 diagnostics-13-00776-f002:**
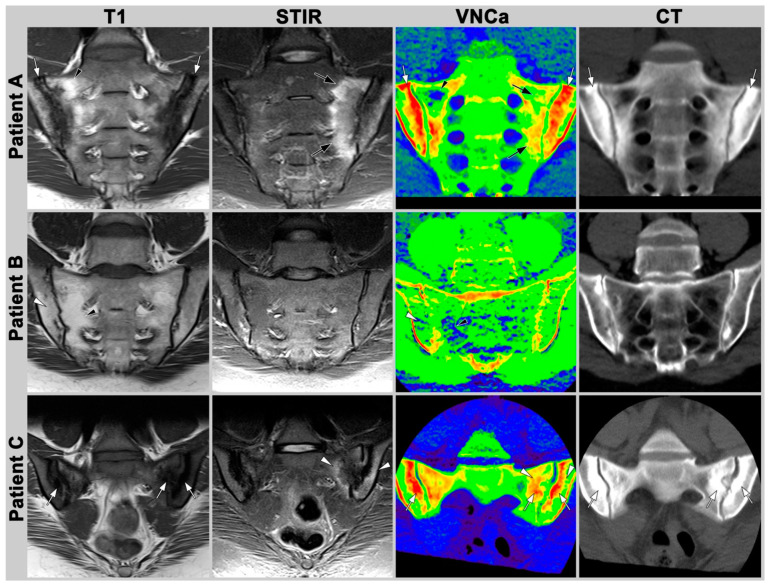
Examples illustrating appearance of osteitis and fatty bone marrow lesions in T1-weighted MR images, short tau inversion recovery (STIR) images, color-coded virtual non-calcium images (VNCa), and standard CT images of the sacroiliac joints.

**Figure 3 diagnostics-13-00776-f003:**
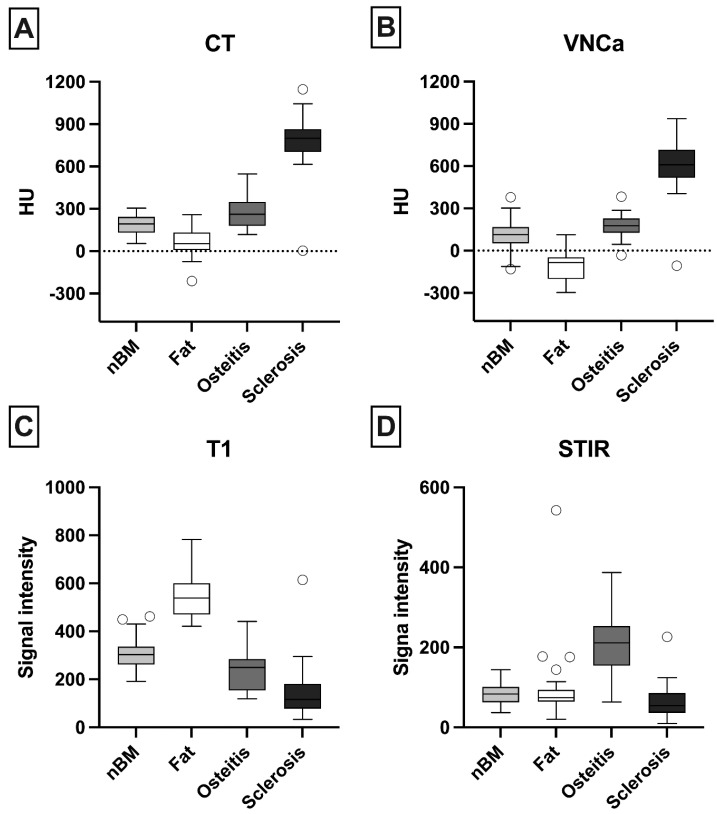
Results of quantitative analysis in (**A**) computed tomography (CT), (**B**): virtual non-Calcium Images (VNCa), (**C**) T1-weighted images and (**D**) short-tau inversion recovery (STIR). Region-of-interest (ROI) measurement was performed in univocal lesions based on subjective scoring. In standard CT, osteitis and fat could not be distinguished from normal bone marrow (nBM). Virtual non-calcium images (VNCa) allowed differentiation between fat metaplasia and normal bone marrow (mean difference: 233.1 HU, *p* < 0.01) but not between osteitis and normal bone marrow (mean difference: −55.67 HU, *p* = 0.27). T1-weighted images showed high signal intensity for fat and were used as gold standard for identification of fat metaplasia, while short-tau inversion recovery (STIR) images showed high signal for osteitis and were used as gold standard for identification of osteitis.

**Table 1 diagnostics-13-00776-t001:** Cross table, sensitivity (SE), specificity (SP), and positive and negative likelihood ratio (LR+/LR−).Values are shown for osteitis and fatty bone marrow lesions on the patient level and on the joint level for both readers together and separately for the expert reader and the beginner scoring dual-energy CT (DECT) using MRI as gold standard. DECT scores show limited diagnostic accuracy for detection of both osteitis and fatty bone marrow deposition. The expert reader (Reader 1) with 13 years of experience in musculoskeletal imaging showed higher performance than the beginner (Reader 2), a research student with 3 years of experience.

Osteitis (Patient Level)	MRI+	MRI-	Overall					Reader 1 (Expert)	Reader 2 (Beginner)
DECT+	11	30	**SE**	73.3%	**LR−**	0.60	**SE**	93.33%	26.67%
DECT+	4	24	**SP**	44.4%	**LR+**	1.32	**SP**	51.85%	70.37%
**Osteitis** (Joint level)	**MRI+**	**MRI-**							
DECT+	14	42	**SE**	46.67%	**LR−**	0.87	**SE**	63.33%	13.33%
DECT-	16	66	**SP**	61.11%	**LR+**	1.2	**SP**	64.81%	81.48%
**Fatty lesions** (Patient level)	**MRI+**	**MRI-**							
DECT+	15	16	**SE**	75%	**LR−**	1.47	**SE**	65%	60%
DECT-	15	33	**SP**	67.3%	**LR+**	0.02	**SP**	77.55%	44.9%
**Fatty lesions** (Joint level)	**MRI+**	**MRI-**							
DECT+	23	19	**SE**	57.50%	**LR−**	0.53	**SE**	50%	47.50%
DECT-	17	79	**SP**	80.61%	**LR+**	2.97	**SP**	83.67%	54.08%

## Data Availability

The data presented in this study are available on request from the corresponding author.

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
