# Peer review of "Dual-Energy-CT for Osteitis and Fat Lesions in Axial Spondyloarthritis: How Feasible Is Low-Dose Scanning?"

_diagnostics, 2023, doi:10.3390/diagnostics13040776_

Round 1

Reviewer 1 Report

DECT bares higher radiation and MRI is a gold standart radiological method to detect axSpA related SIJ changes; the logic of the present study is not clear. 

Author Response

Response to Reviewer 1 Comments

Point 1: DECT bares higher radiation and MRI is a gold standart radiological method to detect axSpA related SIJ changes; the logic of the present study is not clear. 

Response 1: Thank your for your important annotation. We tried to transfer early scientific evidence to a clinical setting by investigating patients with various conditions and reducing the radiation exposure of DECT scans. Thereby, we wanted to analyse the performance of readers with different levels of experience. Our hypothesis was that DECT, performed with a low dose, could be an option for patients with contraindications for MRI. However, in our study, the diagnostic performance of low-dose DECT for osteitis and fatty bone marrow lesions was merely moderate. To clarify the aim of the study, we added:

  1. “Furthermore, CT offers some additional advantages over MRI, for example, lower costs and shorter scanning times which is relevant for patients who cannot undergo MRI-scans due to back pain. Additionally, DECT might be an alternative imaging technique for patients with contraindications for MRI, e.g. pacemaker. However, DECT will use ionizing radiation for imaging.”

  1. “Therefore, the aim of our study was to assess the diagnostic accuracy of low-dose DECT virtual non-calcium (VNCa) images for detecting bone marrow pathologies of the SIJs, i.e. osteitis and fatty bone marrow deposition in patients with axSpA. We wanted to transfer earlier scientific findings to a clinical setting by investigating patients with various conditions”.

  1. “Still, MRI remains gold standard for detection of inflammatory bone marrow lesions and DECT performed as low-dose CT could not achieve diagnostic accuracy of MRI. However, previous studies showed a good diagnostic performance for DECT using higher radiation and offer a possible alternative for patients with contraindications for MRI.”

.

Reviewer 2 Report

Dear Editor, Dear Authors,

The manuscript titled: Dual-Energy-CT for osteitis and fat lesions in axial spondyloarthritis: how feasible is low-dose scanning? (diagnostics-2137568) analyses the ability of ld-DECT virtual non-calcium (VNCa) images for detecting bone marrow pathologies of the SIJs in patients with axSpA. Authors gathered imaging analysis of sixty-eight patients with suspected or proven axSpA. Patients underwent ld-DECT and MRI of the SIJ. VNCa images were reconstructed from DECT data and scored for the presence of osteitis and fat-ty bone marrow deposition by two readers with different experience (beginner and expert). The analysis showed osteitis (n=28) and fatty bone marrow deposition (n=31). Overall correlation with MRI was moderate for osteitis and fatty bone marrow deposition. Authors concluded that ld-DECT failed to detect osteitis or fatty lesions in patients with suspected axSpA and probably higher radiation might be needed for DECT-based bone marrow analysis.

The manuscript is well-written and shows various aspects of radiological imaging. However few questions arise:

1.       Have you seen the difference between analyzed T1, STIR, CT, and VNCa males and females in imaging?

2.       How long last the diseases (particularly inflammatory spondyloarthropathy) last?

3.       How were the patients treated? Did they use biological DMARDs?

If the authors consider the above comments, I believe the work is valuable and can be published in the International Journal of Diagnostics. 

Reviewer 3 Report

This is an excellent piece of work.  For diagnostic purposes, the technique in question (lose dose DECT) did not perform well, particularly because it could not detect osteitis with accuracy, and this was brought out by the investigators.

Author Response

Response to Reviewer 3 Comments

Point 1: This is an excellent piece of work. For diagnostic purposes, the technique in question (lose dose DECT) did not perform well, particularly because it could not detect osteitis with accuracy, and this was brought out by the investigators.

Response 1: Thank you for your comment. Your positive critique is well-appreciated.